# Enhancing iCVD Modification of Electrospun Membranes for Membrane Distillation Using a 3D Printed Scaffold

**DOI:** 10.3390/polym12092074

**Published:** 2020-09-12

**Authors:** Nicole Beauregard, Mustafa Al-Furaiji, Garrett Dias, Matthew Worthington, Aravind Suresh, Ranjan Srivastava, Daniel D. Burkey, Jeffrey R. McCutcheon

**Affiliations:** Department of Chemical and Biomolecular Engineering, University of Connecticut, 191 Auditorium Rd. Unit 3222, Storrs, CT 06269-3222, USA; nicole.beauregard@uconn.edu (N.B.); alfuraiji79@gmail.com (M.A.-F.); garrett.dias@uconn.edu (G.D.); matthew.worthington@uconn.edu (M.W.); asuresh01@manhattan.edu (A.S.); ranjan.srivastava@uconn.edu (R.S.); daniel.burkey@uconn.edu (D.D.B.)

**Keywords:** membrane distillation, initiated chemical vapor deposition, electrospinning, nanofibers, water

## Abstract

Electrospun membranes have shown promise for use in membrane distillation (MD) as they exhibit exceptionally low vapor transport. Their high porosity coupled with the occasional large pore can make them prone to wetting. In this work, initiated chemical vapor deposition (iCVD) is used to modify for electrospun membranes with increased hydrophobicity of the fiber network. To demonstrate conformal coating, we demonstrate the approach on intrinsically hydrophilic electrospun fibers and render the fibers suitable for MD. We enable conformal coating using a unique coating procedure, which provides convective flow of deposited polymers during iCVD. This is made possible by using a 3D printed scaffold, which changed the orientation of the membrane during the coating process. The new coating orientation allows both sides as well as the interior of the membrane to be coated simultaneously and reduced the coating time by a factor of 10 compared to conventional CVD approaches. MD testing confirmed the hydrophobicity of the material as 100% salt rejections were obtained.

## 1. Introduction

Membrane distillation (MD) is a thermally driven desalination process that allows water vapor molecules to travel through a porous hydrophobic membrane [1]. Dissolved solids entrapped in the liquid stream are unable to pass through the membrane. Challenges remain in identifying appropriate membrane materials for MD as commercialization of the technology has stalled. MD membranes must be entirely hydrophobic to prevent any liquid penetration through the membrane while allowing vapor transport and preventing thermal transport. This means that these membranes must exhibit high porosity, low tortuosity, low thermal conductivity, and high mechanical and thermal stability [2]. The combination of these properties, some of which are counter to one another (for example, high porosity with good mechanical properties), leave limited options for designing MD membranes in commercial applications.

MD membranes are commonly made from hydrophobic polymers such as polytetrafluoroethylene (PTFE), polypropylene (PP), and polyvinylidene fluoride (PVDF). PTFE is a hydrophobic polymer and has shown good salt rejection and water flux in MD [3]. PTFE is, however, challenging to process and is only made into membranes through a costly sintering or melt extrusion process [4,5]. Polypropylene is a lower-cost alternative to PTFE, but it lacks the same thermal stability [6]. PVDF is less hydrophobic than PTFE, but is also less expensive and easier to process with conventional solvents. This allows it to be used in a wider variety of polymer processing techniques rather than traditional sintering, melt-extrusion or phase inversion. New polymer processing techniques can offer more of the desired MD properties than the traditional fabrication techniques.

Electrospinning has been shown to be successful as a means of fabricating MD membranes [7]. In this technique, a polymer solution is extruded through a needle supplied with a voltage. The resulting fiber is collected in a random fashion on a grounded collector surface. These membranes exhibit many desirable MD characteristics including high porosity and high strength-to-weight ratio [8]. Other properties, including fiber diameter and thickness, can be easily tuned by changing the solution and spinning parameters. This fabrication process is limited, however, by the scarcity of easy-to-spin hydrophobic polymers that can be made into ultra-fine fibers (for example less than 500 nm in diameter). Spinnable hydrophobic polymers include PVDF, polystyrene, and polyimide [9,10,11]. Electrospun PVDF membranes have already shown good promise for use in MD [10,12]. PVDF was shown to exhibit a higher water flux than commercial, non-electrospun, PVDF membranes [13]. If hydrophilic polymers could be used, we could expand the list of spinnable polymers that have been shown to create ultra-fine fibers (such as polyacrylonitrile and nylon) or other desirable characteristics. If used, mats made from these polymers would require some type of surface modification to render them hydrophobic. It is also conceivable that modifying hydrophobic fibers with more hydrophobic materials would improve performance.

Any modification of a fiber mat must yield conformally coated fibers so as to prevent wetting of the membrane. Furthermore, coating would preferably not change the structural properties of the mat (such as fiber size or mat porosity). Other studies describing the modification of MD membranes to increase their hydrophobicity have resulted in a decrease in flux [14,15,16]. This was attributed largely to the decrease of pore size and porosity.

Initiated chemical vapor deposition (iCVD) has emerged as a modification technique for porous materials that can produce an ultra-thin, uniform and defect-free coating, which retains the structural morphology of porous materials [17,18,19,20]. iCVD polymer films are formed via free radical polymerization [21,22] where a monomer of the desired polymer and an initiator are volatized and delivered into the deposition chamber at low pressure. The initiator then encounters heated filaments that decompose the initiator into radicals. The radicals and monomers are then adsorbed onto the substrate, which has traditionally been fixed to the bottom of the chamber. The bottom of the chamber is cooled to enhance adsorption [23,24]. Once adsorbed, free radical polymerization occurs on the surface forming the thin polymeric film. The solvent-free nature of this process allows for a uniform, conformal coating of high aspect ratio substrates without altering surface morphology [25,26]. For electrospun membranes, this ideally results in a conformal coating of the individual fibers without significantly altering fiber size, morphology, or mat porosity. Solvents can introduce potential impurities into the coating and can possibly react with the monomer forming undesirable side products. iCVD is suitable for fragile or heat-sensitive substrates as the substrate is not directly exposed to high temperatures or mechanical stress during the process. iCVD has also been used to increase the hydrophobicity of hydrophilic substrates in other studies [26,27]. In previous work, 1H,1H,2H,2H-perfluorodecylacrylate (PFDA) has successfully been used to render hydrophilic electrospun membranes suitable for MD, however this polymer cannot be used commercially due to environmental concerns [28]. Divinylbenzene (DVB) has emerged as a more environmentally friendly alternative, but the deposition time required to achieve suitable hydrophobicity is much longer [29]. Shortening the time necessary for deposition would increase the likelihood that iCVD could be used for membrane modification.

This work aims to investigate a new method for iCVD coating porous media (and nanofibers in particular) in order to expedite coating and reduce the amount of reactants by harnessing convective flow during deposition. Conventional iCVD is reliant on diffusion of the monomer to the membrane, which is affixed to the bottom of the chamber. In order to achieve adequate hydrophobicity using this approach, both sides of the membrane must be coated individually in subsequent coating sequences. By using a porous scaffold that is oriented in such a way that the monomers and radicals must past through the mat prior to exiting the chamber, we create a convective flow of monomers through the mat, which increases deposition throughout the mat thickness. This has the added benefit of improving the conformal nature of the modification while also accelerating deposition. The success of using this approach will be verified by coating hydrophilic electrospun nanofibers with a hydrophobic polymer and investigating its performance in membrane distillation. Hydrophilic fibers are used to both demonstrate the expanded material options for MD membranes after modification but also to verify that the modification is in fact conformal and defect free.

## 2. Experimental

### 2.1. Materials

Polyacrylonitrile (PAN, MW = 150,000) was purchased from Scientific Polymer Products (Ontario, NY, USA). N, N-dimethylformamide (DMF, anhydrous, 99.8%) was purchased from Acros Organics (Morris Plains, NJ, USA). Sodium chloride (NaCl, crystalline, certified ACS) was obtained from Fisher Scientific (Pittsburgh, PA, USA). Divinylbenzene (DVB, Technical Grade, 80%) and *tert* butyl peroxide (TBPO, 98%) were used as purchased with no further purification from Sigma Aldrich (St. Louis, MO, USA). 

### 2.2. Electrospinning

PAN nanofiber mats were produced in a custom-built electrospinning chamber. To prepare the 8% PAN solution, PAN polymer powder was dissolved in DMF and stirred for 24 h at 60 °C. Of the polymer solution, 20 mL was spun and collected on a rotating drum covered with aluminum foil at a rate of 1.5 mL/h. The voltage was maintained between 22 and 28 kV and the relative humidity in the chamber was maintained around 20%. To prepare the 8% PVDF solution, PVDF polymer powder was dissolved in a 60:40 ratio of DMF and acetone. The solution was stirred for 24 h at 60 °C. Of solution, 20 mL was electrospun onto a fabric sheet at a rate of 4 mL/h. The voltage was between 22 and 28 kV and the relative humidity was maintained at 40%. The PVDF membrane will serve as a hydrophobic control to compare with the iCVD coated hydrophilic PAN membrane.

### 2.3. Scaffold Fabrication

A schematic of the 3D printed scaffold was first designed using SOLIDWORKS. The 3D printed scaffold was designed to suspend the membrane in a new orientation normal to the flow of reactants into the chamber. The scaffold was devised to harness the effect of convective flow of the reactants to force them to diffuse through the membrane to reach the vacuum pump on the other side of the scaffold. The only path for the reactants to the vacuum would be through the porous membrane, where they would deposit throughout the structure. This approach eliminates the need for flipping the membrane over for a second coat on the back side, saving time and reactants. 

The membrane was mounted vertically on a mesh in the middle of the scaffold as shown in Figure 1 and Figure 2. A removable membrane stage was designed to hold and support the membrane during the coating. The device itself conformed to the geometry of the chamber allowing for the porous membrane to be the most direct path for the monomers to exit the chamber. Two supports for the membrane were fabricated that can be slid into the membrane stage depicted in Figure 1. A mesh support was fabricated to allow for convective flow through the membrane. A solid blank support was also fabricated with no cutouts, which should prevent convective flow through membranes as in the traditional case of the membrane fixed to the stage. Caps were included on top of the scaffold to ensure the two pieces held together during the process and to minimize the air gap on top of the scaffold (which could cause bypassing of the membrane).

The scaffold was fabricated using a Form 1+ 3D printer (Formlabs, Somerville, MA, USA). This printer uses stereolithography where a photo-reactive liquid resin is cured via a system of lasers and mirrors causing localized photopolymerization [30]. Figure 1 also shows the completed 3D printed scaffold. The membrane was mounted and dismounted from the mesh without deformation.

### 2.4. Initiated Chemical Vapor Deposition

iCVD experiments were conducted in a custom-built reactor (GVD Corporation, Cambridge, MA, USA). The initiator, tert-butyl peroxide (TBPO), was delivered into the chamber in the vapor phase at rate of 15.0 sccm using a mass flow controller (MKS 1479, Andover, MA, USA). TBPO was held at ambient temperature due to its high volatility. The monomer, DVB, was heated to 70 °C and delivered in the chamber at a rate of 3.0 sccm through a delivery line heated at 80 °C to prevent condensation. To prevent auto-polymerization, CuCl_2_ powder was added to the vessel containing liquid DVB. A vacuum pump (Edwards E2M40, Chelmsford, MA, USA), pressure transducer (MKS 622, Andover, MA, USA), and throttle valve (MKS 153D, Andover, MA, USA) were used to maintain the pressure in the chamber at 150 mTorr. The temperature of the stage was controlled using a water-cooled chiller held at 12 °C. Electrically heated nichrome filaments (Omega NI80-111 020, Norwalk, CT, USA) were suspended above the cooled stage and were maintained at 300 °C. 

Experiments were conducted using the standard iCVD approach and the scaffold coating approach. In the standard process, the membrane was affixed to the cooled stage at the bottom of the chamber. Deposition occurred as the reactants diffuse to the bottom and adhere onto the exposed fibers. The majority of the reactants did not polymerize and were expelled out of the chamber through a vacuum pump. Adsorption was found to be enhanced at lower substrate temperatures [24]. Higher deposition rates were reported for membranes held at lower stage temperatures. The method relies on temperature as the rate-determining parameter. In order to achieve adequate hydrophobicity, the membrane must be coated on each side separately, which adds additional processing time as the chamber must be purged and then brought back under vacuum again. There is a low conversion rate for the polymerization and the much of the reactants are wasted [31].

In the new approach, the membrane is held in place through the use of the scaffold at a position normal to the flow of reactants into the chamber. This position allows for convective flow of reactants through the membrane rather than relying strictly on diffusion. Convective flow through the membrane allows for another mechanism for coating to occur on the membrane fibers that could potentially coat the backside of the membrane eliminating the need for two rounds of coating.

The introduction of the scaffold required some chamber modification. The scaffold was set approximately in the center of the chamber effectively dividing the chamber into two halves. The filament array was only strung across the half of the chamber before the scaffold where the reactants enter and the half facing the vacuum pump was empty. To ensure deposition was occurring, a silicon wafer was placed beneath the filaments and laser interferometry was used to measure the thickness of the polymer film. The membrane was affixed to a support, which can be slid into the membrane stage. The membrane stage was then slid into place in the scaffold to suspend the membrane above the cooled stage. Deposition time was monitored and tests were run at intervals between 15 and 200 min to study the effect of coating time on the hydrophobicity of both sides of the membrane.

Previous work on iCVD has shown that the monomer deposition rate is strongly influenced by substrate temperature and that lower substrate temperatures yield higher deposition rates [24]. In the scaffold orientation, there is no temperature control as only the stage can be cooled. In order to eliminate temperature as a variable and focus solely on the effect of convective flow, experiments were conducted with the membrane held in place by the mesh support and subsequently the solid blank support. The non-convective blank support should prevent flow through the membrane mimicking how the membrane is coated using the standard method while eliminating temperature as a variable. This offers better insight into the role of convective flow in coating the membrane.

### 2.5. Membrane Distillation

DVB coated electrospun PAN and uncoated electrospun PVDF membranes, and a Millipore PVDF membrane (effective area of 3 × 1 inch^2^) were tested in a direct contact membrane distillation (DCMD) benchtop system. The operating conditions were identical for each membrane. The feed solution was a 5 M NaCl solution and DI water was used as the permeate solution. A 5 M feed solution was used for easy detection of leaks and wetting in a DCMD test. A 30 °C temperature difference was maintained between the two streams, where the feed was kept at 50 °C and the permeate at 20 °C. The permeate flux was measured over the 6 h duration of the experiment by measuring the change in weight of the permeate tank. Salt rejection was calculated by measuring the conductivity of the permeate tank. Experiments were performed in triplicate for each membrane type.

The performance metrics used to characterize the performance of the membranes in DCMD are water flux and salt rejection percentage. Water flux is the flow rate of water through the membrane per unit area. This quantity is measured gravimetrically by recording the weight change of the permeate tank over time using a balance and is measured in units of LMH. It is calculated by Equation (1):(1)$$J_{w}=\frac{\Delta w}{\rho At}$$

Salt rejection is the percentage of solute rejected by the membrane. It can be calculated by Equation (2):(2)$$R=\left(1-\frac{C_{p}}{C_{f,i}}\right)\times 100$$
where conductivity measurements were used to measure salt concentration in the permeate.

### 2.6. X-ray Photoelectron Spectroscopy (XPS)

X-ray photoelectron spectroscopy (XPS) was carried out on the DVB coated samples to confirm deposition of the polymer by iCVD. For comparison, spectra were also obtained from an uncoated PAN membrane, and a membrane exposed only to the initiator and not the monomer through iCVD. A PHI multiprobe system with an Al source was used. Survey XPS were obtained between binding energies of 0 and 1100 eV, with 250 W power, 100 eV pass energy, 1 eV resolution, and 50 ms dwell time. The spectrum collected for each sample was an average of 60 scans.

## 3. Results and Discussion

### 3.1. Membrane Characterization

Membrane distillation experiments were first carried out with PVDF membranes to generate benchmark performance data for comparison to electrospun membranes modified using iCVD. One membrane was a commercial 0.45 µm non-electrospun commercial membrane. The other membrane was an electrospun PVDF membrane fabricated using the same procedure as the PAN membranes. Both PVDF membranes exhibited 100% salt rejection without any modification, reflecting the inherent hydrophobicity of PVDF. We note that the electrospun membrane exhibited a significantly higher water flux, but demonstrating this is not the purpose of this work.

Electrospun mats were then coated with DVB using iCVD. Membranes were first coated using the standard iCVD approach of affixing the membrane to the bottom of the chamber. It was observed that the standard protocol for coating DVB on PAN membranes required 100 min of deposition on each side of the membrane (200 min total) to achieve adequate MD hydrophobicity. The permeate flux was observed to be approximately 12 LMH after 6 h of operation with 99.99% salt rejection. In the subsequent experiments where the scaffold was implemented, the deposition was conducted on only one side of the membrane to determine if the new coating method could coat both sides simultaneously.

SEM pictures were taken of both the uncoated and coated electrospun membrane samples as shown in Figure 3. There was no identifiable change in nanofiber morphology between the two samples. While this does not prove that the membrane has been conformally coated, it does imply that the coating process does not cause any damage or noticeable change to the membrane structure (such as clogging or a change in effective pore size) that could impact its performance. The pores appear unobstructed, which should allow for water flux to be observed if the membrane does not wet out during the process.

XPS was conducted on an uncoated PAN membrane, a membrane exposed to only the initiator (PAN-TBPO), and a DVB coated membrane (PAN-PDVB) to verify the presence of an iCVD coating. Figure 4 shows XPS spectra obtained from PAN, PAN-TBPO, and PAN-PDVB membranes. The as-spun PAN membrane (top) had a prominent N1s peak, due to the nitrile functionality. This N1s peak exhibited lower intensity with the PAN-TBPO membranes, suggesting some chemical modification of the surface of the PAN membrane upon exposure to TBPO alone. As TBPO is a radical generator, interactions of the radicals with the membrane surface upon heating in the iCVD system can potentially cause chemical modifications of the surface. This is likely, as evidenced by the increase in the O1s peak in the PAN-TBPO sample (middle spectrum). Abstraction of surface hydrogen on the membrane by TBPO radicals would result in oxygen and adventitious water uptake by the membrane upon exposure to atmospheric conditions post-deposition. The N1s peak completely disappeared in PAN-PDVB (bottom spectrum), confirming the presence of a PDVB layer devoid of nitrogen around the PAN nanofibers. The thickness of the layer was large enough (>10 nm) to suppress the N1s signal from the PAN nanofibers underneath but thin enough to not be apparent using SEM. The O1s peak in the PAN-PDVB sample was also increased over PAN alone, which could be due to the reasons mentioned above for the PAN-TBPO case and additionally due to some incorporation of the TBPO end groups into the PDVB polymer.

Contact angle measurements were conducted on both sides of the membrane for standard coated membranes (with the membrane on the bottom of the chamber) and those mounted on the scaffold with and without convective flow. Scaffold mounted membranes were tested without convective flow (as a control) by replacing the mesh with solid piece that prevented any flow through the membrane. Results shown in Table 1. All membranes coated, including that which was coated conventionally, exhibited hydrophobicity on the side of the membrane facing the monomers. The membranes coated on the scaffold with convective flow and the longest coating times had the highest contact angles on both sides of the membranes. Testing the back of the membrane was used as an indicator of conformal coating. Longer coating times (30 and 50 min) and convective flow both led to higher contact angles on the back of the membrane. Scaffold coated membranes without convective flow coated for only 15 min wetted out on the back side, indicating that the coating was incomplete. This is part of the value of using a hydrophilic fiber to determine coating conformity: any failure to coat the fibers will result in wetting. 

### 3.2. Membrane Distillation Tests

It was observed that the membranes coated between 20 and 200 min all exhibited 100% salt rejection in MD indicating no wetting during the test. Long term flux stability data is shown in Figure 5 for a membrane coated for 20 min via convective flow. The membrane exhibited 100% salt rejection and a stable flux over the duration of the experiment.

Figure 6 displays the flux results for a variety of coating times and coating orientations. The membranes exhibited reasonable water flux relative to our commercial control and PVDF nanofiber control. We noted some improvement in water flux with decreasing coating times, though these results were not definitive. We did note that the very long coating times allow for more deposition to occur on the fibers, which may decrease the pore size, though this was not observable under SEM.

While flux and rejection performance were not improved over conventional hydrophilic polymers, we did find that our convective coating technique could effectively turn a hydrophilic membrane into a membrane that does not get wet in DCMD. We achieved this while eliminating the need for two rounds of coating as required in the standard method and thus drastically lessened the time needed to prepare the modified membrane. The improved contact angle results also demonstrated the scaffold’s ability to conformally coat porous materials. This was verified by the 100% salt rejections exhibited by all of the membranes described in Figure 6.

Not included in Figure 6 is data from our 15 min coating tests. While these membranes initially produced reasonable fluxes and rejection, the membranes lost flux and salt rejection during the 6 h experiment, which indicated wetting was taking place. We deemed that using our convective flow method, a minimum 20 min of coating time was required to prevent wetting.

In general, though, coating time reduction was notable in this work. The previous shortest coating time was 35 min on each side (70 total plus changeover and startup time for the second coating time) to achieve a membrane with no wetting) [29]. Using our scaffold method, the coating time can be reduced to a single 20 min run. Not only does this save time, but it saves reactants as most of the monomers that pass through the chamber are vented using the conventional iCVD approach.

## 4. Conclusions

This work introduced a novel and rapid technique for conformally coating porous materials using iCVD. Instead of relying on the natural diffusion of the reactants, the innovative design of our scaffold harnesses the convective flow of the through the membrane. This unique configuration yields improved coating of the back side of the membrane not in direct contact with the precursors, which previously required an additional round of coating. The effectiveness of the technique was demonstrated by modifying a hydrophilic material that would normally wet spontaneously when exposed to water.

The benefits of this new method include improved hydrophobicity on both sides of the membrane, faster overall coating, and reduced process dead time. Coating time could also be further reduced by introducing a cooling component into the scaffold design. This could be achieved by either adding a cooling loop to the scaffold or fabricating the device out of metal to allow for heat transfer from the scaffold to the cooled stage. Considering the MD results for membranes coating using the standard method, temperature did play a role in the coating process, and higher deposition rates were observed with lower substrate temperatures.

## Figures and Tables

**Figure 1 polymers-12-02074-f001:**
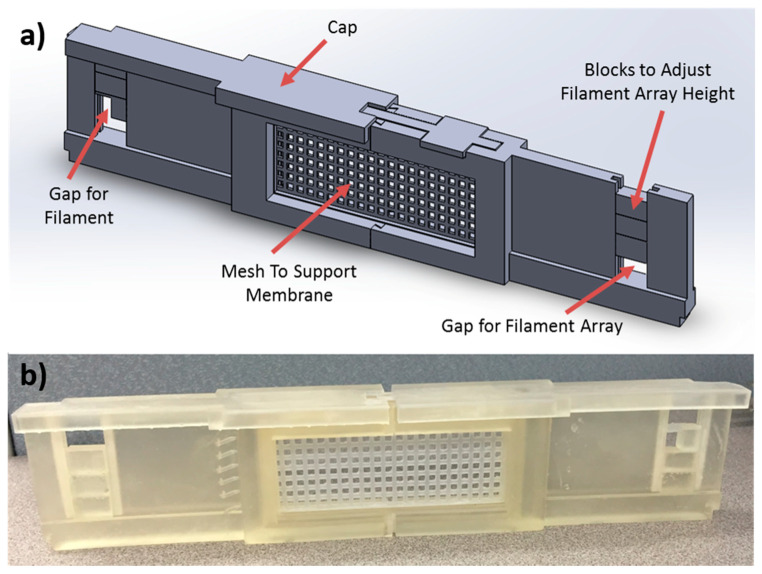
(**a**) Solidworks schematic of the support and (**b**) 3D printed support. The porous structure could be replaced with a solid, impermeable plate to serve as a control. Please note that this figure is best viewed in color.

**Figure 2 polymers-12-02074-f002:**
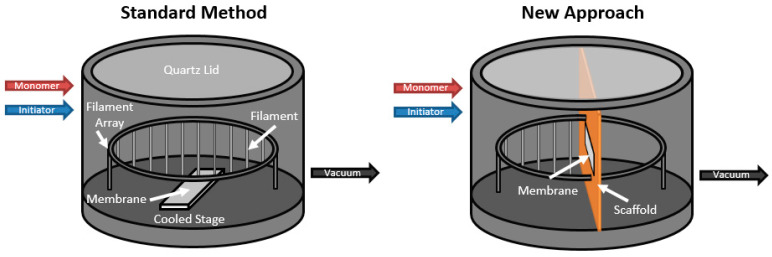
Schematics of the initiated chemical vapor deposition (iCVD) chamber demonstrating the standard method (**left**) and the new approach using the scaffold (**right**).

**Figure 3 polymers-12-02074-f003:**
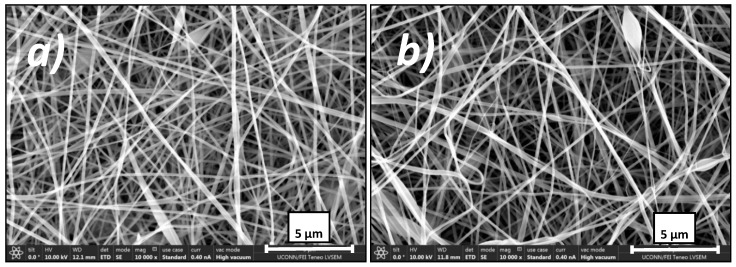
SEM images at 10,000× magnification of (**a**) an uncoated electrospun PAN membrane and (**b**) a coated electrospun PAN membrane modified via the scaffold iCVD method. Note the scale bar of 5 μm.

**Figure 4 polymers-12-02074-f004:**
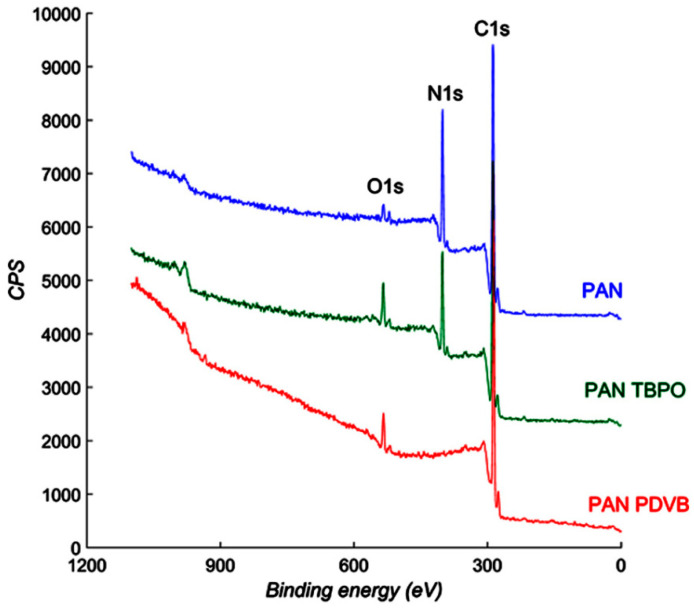
XPS collected from PAN, PAN-TBPO, and PAN-PDVB samples.

**Figure 5 polymers-12-02074-f005:**
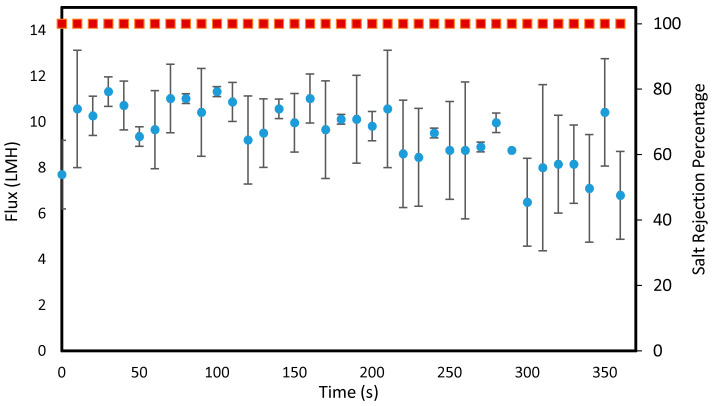
Water flux performance over six hours for a PAN membrane modified in iCVD using the convective mesh support for 20 min. Experimental conditions: feed—5 M NaCl solution at 50 °C, permeate—DI water at 20 °C, flowrates—0.4 GPM, and no pressure difference.

**Figure 6 polymers-12-02074-f006:**
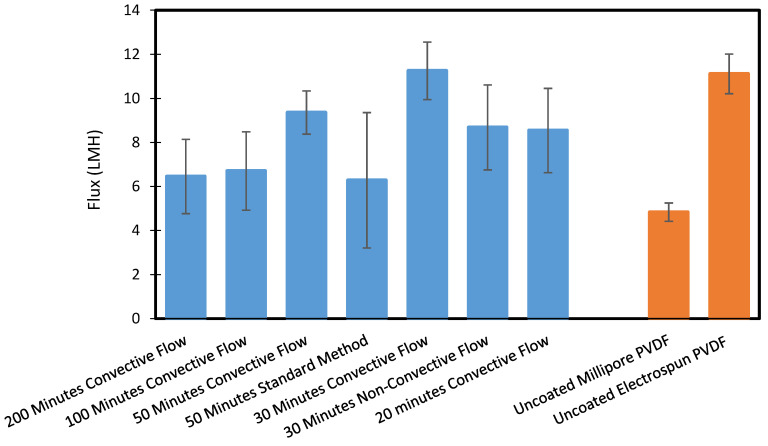
Water flux comparison between iCVD deposition times of pDVB-coated PAN membranes using the 3D printed scaffold, a commercial Millipore PVDF membrane, and an electrospun PVDF membrane. The membranes were tested in direct contact membrane distillation (DCMD) for 6 h and flux values were averaged over that time. All experiments yielded a 100% rejection, which indicated no wetting occurred during the test. Experimental conditions: feed—5 M NaCl solution at 50 °C, permeate—DI water at 20 °C, flowrates—0.4 GPM, and no pressure difference.

**Table 1 polymers-12-02074-t001:** Contact angle measurements for selected coating orientations and coating times of both sides of iCVD coated electrospun PAN membranes.

Coating Orientation	Coating Time (Minutes)	Contact Angle of Side Directly Exposed to Precursors	Contact Angle of Side Not in Direct Contact with Precursors
Standard: Affixed to Stage	50	122 ± 4.3°	118 ± 4.0°
Scaffold: Allowing Convective Flow	50	135 ± 4.8°	147 ± 4.3°
Scaffold: Allowing Convective Flow	30	149 ± 3.4°	141 ± 1.5°
Scaffold: Prohibiting Convective Flow	30	135 ± 0.9°	127 ± 0.7°
Scaffold: Allowing Convective Flow	15	136 ± 0.6°	125 ± 0.4°
Scaffold: Prohibiting Convective Flow	15	124 ± 0.5°	NA

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
