# Peer review of "Enhancing iCVD Modification of Electrospun Membranes for Membrane Distillation Using a 3D Printed Scaffold"

_polymers, 2020, doi:10.3390/polym12092074_

Round 1

Reviewer 1 Report

The paper focuses on the development of a new method for iCVD coating porous media in order to expedite coating and reduce the amount of reactants by harnessing convective flow during deposition.

please add a comment on electrospun mat porosity in comparison of the commercial structure and the impact of the coating on this. Please add also the fieber dimension. are the fiber unidorm in the mats? only a small part could be appreciated from SEM images. is the process sustainable?

Author Response

please add a comment on electrospun mat porosity in comparison of the commercial structure and the impact of the coating on this.

Response:

Thank you for the comment. The commercial membrane tested was already hydrophobic and no coating was applied to this membrane.

We did not measure the porosity of the two materials since it only constitutes one feature of the material which impact water flux.  Nanofiber mats are well known to have high porosity and interconnectivity relative to phase inversion cast membranes

Please add also the fieber dimension. are the fiber unidorm in the mats? only a small part could be appreciated from SEM images.

Response:

The reviewer makes a good point. The fibers are uniform in the mats. Our SEM images were representative of the mat images that we collected at large. We increased the size of the scale bar to better display the fiber size. Our previous published work for electrospinning calculated the average diameter for the conditions and polymer we used to range from 0.19-0.43 μm uncoated which is comparable to the coated fibers. We added the following sentence to the manuscript:

“Previous work determined the size of the PAN nanofibers spun under these conditions ranges from 0.19-0.43 μm”

is the process sustainable?

Response:

It is unclear what the reviewer means by “sustainable”.  MD is a process that may be sustainable for the concentration of brines.  Electrospinning is a membrane fabrication process that is as “sustainable” as any casting process (since it is a solution casting process in and of itself). CVD is a modification process, but this paper is not intended to comment on it’s “sustainability”.  Our process does, however, better utilize the CVD monomers by ensuring that as much of the injected monomer in the chamber has the opportunity to coat the membrane.

Reviewer 2 Report

The manuscript shows the combination of several advanced polymer treatment methods, i.e. initiated chemical vapor deposition (iCVD), electrospinning, and also 3D printing. These methods are integrated for increasing the hydrophobicity of electrospun membranes, and thus for a potential better application in water distillation. The results about salt rejections are positive. These contents are interesting. I have no hesitation to recommend its acceptance for publication in POLYMERS. However, the writing still has rooms to be improved. Some of my suggestions for the authors’ references are listed as follows:

In the INTRODUCTION section, it should be better to give several sentences about the most recent development of electrospinning, which can both project the merits of your work, and also provoke the readers’ interests in future, and thus increase the impact of your paper after publications, e.g. electrospinning is extending its capability of creating novel functional nanomaterials along two ways. One is the multifluid processes for generating core-shell (10.1016/j.rinp.2019.102770), Janus (Polymers, 2020, 12, 103), tri-layer core-shell (10.1016/j.carbpol.2020.116477), and other complicated nanostructures (10.1002/wnan.1601) and also for treating more types of working fluids (Polymers 2019,11,1287). The other is the combination of electrospinning with other traditional/advanced materials processing methods (10.1016/j.matdes.2020.108782). Your work is just an excellent example of the second direction.     

The information about the raw PVDF including the manufacture and molecular weight should be provided in the “2.1 Materials” section.

The scale bars in Figure 3 are too small and please enlarge for the readers.

The references’ formats are chaotic, lacking journal names, upper cases of all the article title words, and so on.  

The references within the most recent three years are too small. It is better to be revised to relate itself within the most recent developments to some extents, particularly for the fast developing fields.

Author Response

The manuscript shows the combination of several advanced polymer treatment methods, i.e. initiated chemical vapor deposition (iCVD), electrospinning, and also 3D printing. These methods are integrated for increasing the hydrophobicity of electrospun membranes, and thus for a potential better application in water distillation. The results about salt rejections are positive. These contents are interesting. I have no hesitation to recommend its acceptance for publication in POLYMERS. However, the writing still has rooms to be improved. Some of my suggestions for the authors’ references are listed as follows:

In the INTRODUCTION section, it should be better to give several sentences about the most recent development of electrospinning, which can both project the merits of your work, and also provoke the readers’ interests in future, and thus increase the impact of your paper after publications, e.g. electrospinning is extending its capability of creating novel functional nanomaterials along two ways. One is the multifluid processes for generating core-shell (10.1016/j.rinp.2019.102770), Janus (Polymers, 2020, 12, 103), tri-layer core-shell (10.1016/j.carbpol.2020.116477), and other complicated nanostructures (10.1002/wnan.1601) and also for treating more types of working fluids (Polymers 2019,11,1287). The other is the combination of electrospinning with other traditional/advanced materials processing methods (10.1016/j.matdes.2020.108782). Your work is just an excellent example of the second direction.    

Response:

Thank you for the thoughtful comment. The idea of triaxial spinning for MD combining hydrophobic and hydrophilic materials could be an interesting idea for further research. The following sentences were added to reflect the recent that work in electrospinning and future potential for MD applications.

“The additional sente Recent electrospinning work has studied coaxial and triaxial electrospinning techniques which could be further investigated for MD application combining both hydrophilic and hydrophobic polymers. [15,16]”

The information about the raw PVDF including the manufacture and molecular weight should be provided in the “2.1 Materials” section.

Response:

The actual molecular weight of this PVDF was not shared by the company. It is described as an ultra-high molecular weight polymer and we do provide the brand and designation. The following sentence was added to the materials section:

 “An ultra-high molecular weight PVDF (Kynar HSV900) was obtained from Arkema Inc. (Kynar is a registered trademark of Arkema Inc.).”

The scale bars in Figure 3 are too small and please enlarge for the readers.

 Response:

We agree that the scale bar was difficult to read in that size. The scale bars in Figure 3 were enlarged along with the font size. 

The references’ formats are chaotic, lacking journal names, upper cases of all the article title words, and so on. 

 Response:

We agree that the citation formats were inconsistent. The format of the references was corrected for clarity and uniformity.

The references within the most recent three years are too small. It is better to be revised to relate itself within the most recent developments to some extents, particularly for the fast developing fields.

 Response:

Additional references 17-22 were added to reflect nanofiber MD work that has been done recently.

“MD performance can also be enhanced via the addition of certain nanomaterials and by fabricating multilayer membranes [17–19].”

“One study combined electrospinning with electrospraying to coat a hydrophobic membrane with an additional hydrophobic aerogel coating to see improved performance [20]. Nanofiber membranes have also been modified for increased hydrophobicity and improved MD performance via surface segregation additives [21]. Additionally, carbon nanotube coatings have been investigated to produce superhydrophobic membranes [22].”